palaeontology/evolution

Oviraptoridae, Late Cretaceous, Theropoda, digit reduction, forelimb evolution

**Author for correspondence:**
Gregory F. Funston
e-mail: gregory.funston@ed.ac.uk

# A new two-fingered dinosaur sheds light on the radiation of Oviraptorosauria

Gregory F. Funston[1,2], Tsogtbaatar Chinzorig[3,4], Khishigjav Tsogtbaatar[4], Yoshitsugu Kobayashi[3], Corwin Sullivan[2,5] and Philip J. Currie[2]

[1]School of GeoSciences, University of Edinburgh, Edinburgh, UK
[2]Department of Biological Sciences, University of Alberta, Edmonton, Alberta, Canada
[3]Hokkaido University Museum, Hokkaido University, Sapporo, Japan
[4]Institute of Paleontology, Mongolian Academy of Sciences, Ulaanbaatar, Mongolia
[5]Philip J. Currie Dinosaur Museum, Wembley, Alberta, Canada

GFF, 0000-0003-3430-4398; CS, 0000-0002-5488-6797

Late Cretaceous trends in Asian dinosaur diversity are poorly understood, but recent discoveries have documented a radiation of oviraptorosaur theropods in China and Mongolia. However, little work has addressed the factors that facilitated this diversification. A new oviraptorid from the Late Cretaceous of Mongolia sheds light on the evolution of the forelimb, which appears to have played a role in the radiation of oviraptorosaurs. Surprisingly, the reduced arm has only two functional digits, highlighting a previously unrecognized occurrence of digit loss in theropods. Phylogenetic analysis shows that the onset of this reduction coincides with the radiation of heyuannine oviraptorids, following dispersal from southern China into the Gobi region. This suggests expansion into a new niche in the Gobi region, which relied less on the elongate, grasping forelimbs inherited by oviraptorosaurs. Variation in forelimb length and manus morphology provides another example of niche partitioning in oviraptorosaurs, which may have made possible their incredible diversity in the latest Cretaceous of Asia.

## 1. Introduction

Oviraptorosaurs are theropod dinosaurs known from an excellent fossil record spanning much of the Cretaceous of Asia and North America [1]. Revived interest in oviraptorosaurs since the 1990s has resulted in a wave of new discoveries, and they are now among the best-known theropods. Aspects of their integument [2,3], reproduction [4–7] and functional morphology [8–11] are

well studied, providing information that is important in understanding the biological changes that accompanied the transition to birds.

Like extant birds, oviraptorosaurs had pennaceous feathers [2,3,12], and most were completely edentulous [1], presumably possessing a keratinous rhamphotheca. They retained two functional oviducts [13], but brooded their eggs like birds [6]. Four main clades of oviraptorosaurs are recognized: the basal caudipterygids, and the more specialized avimimids, caenagnathids and oviraptorids. Of these, oviraptorids are known from the best material and are the most speciose, but they are restricted to China and Mongolia. Between the Nanxiong Formation of China and the Nemegt Basin of Mongolia, at least 15 oviraptorid genera are known, of which eight have been described in the last decade [14–16]. This flurry of discovery has documented one of the last diversifications of non-avian theropods prior to the Cretaceous–Palaeogene (K-Pg) extinction [14,15].

Despite this rich record, it is unclear why oviraptorids radiated during the late Campanian–Maastrichtian, when the diversity of other theropod groups remained stable [17]. This is partly because there is little consensus on relationships within the main oviraptorosaur clades, but also because the rapid rate of discovery has outpaced macroevolutionary analyses. Regardless, this radiation is important given that patterns of dinosaur diversity preceding the K-Pg extinction are debated, and perceived decrease in richness [18,19] and disparity [17] during the Maastrichtian may be the result of undue extrapolation from the well-studied North American fossil record [20,21]. In North America, most groups of dinosaurs reach a diversity peak in the Campanian [19,22,23], followed by stability [20,23] or decrease [17–19] in the Maastrichtian. Diversity trends in Asia are less well known, but there is evidence of stability in most groups except hadrosaurs [17], which become increasingly disparate towards the Maastrichtian. The radiation of oviraptorids throughout the Campanian–Maastrichtian provides another line of evidence that diversity patterns in North America may not be representative of global trends.

Here, we describe a bizarre new oviraptorid from the Maastrichtian Nemegt Formation of Mongolia, with a reduced, functionally didactyl forelimb. The new taxon, *Oksoko avarsan* gen. et sp. nov., known from multiple associated skeletons, represents the sixth genus of oviraptorid and ninth genus of oviraptorosaur from the Nemegt Formation, adding to previous evidence for a remarkable diversity of oviraptorosaurs in the Maastrichtian of Asia. In addition to revealing unambiguous gregariousness in oviraptorids, the new taxon sheds light on their radiation in the Late Cretaceous. *Oksoko avarsan* increases the already considerable range of known variation in the lengths and morphologies of the forelimb and manual digits among oviraptorids, which in turn suggests functional variation that might be related to foraging, nesting, display or other behaviours. Ancestral state reconstruction based on a revised phylogeny shows that forelimb and manual digit reduction occurred in the single oviraptorid clade Heyuanninae, coinciding with heyuannine dispersal from their ancestral range in southern China to what is now the Gobi Desert. The conjunction of forelimb reduction and biogeographic dispersal suggests the expansion of heyuannines into a new niche at the end of the Cretaceous.

## 2. Results

Theropoda Marsh 1881 [24].

   Oviraptorosauria Barsbold 1976a [25].

   Oviraptoridae Barsbold 1976b [26].

   Heyuanninae (=Ingeniinae) Barsbold 1981 [27].

   *Oksoko avarsan* gen. et sp. nov. (figures 1–3).

   **Etymology**. *Oksoko* (pronounced 'Oak-soak-oh') from the three-headed eagle of Altaic mythology, in reference to the fact that the holotype assemblage preserves three skulls; the specific name *avarsan* is from the Mongolian word 'аварсан' (avarsan: rescued), reflecting their confiscation from poachers and/or smugglers.

   **Holotype**. Institute of Paleontology, Mongolia (MPC-D) 102/110a, a nearly complete juvenile skeleton missing only the distal half of the tail (figures 1–3), preserved in an assemblage of four individuals.

   **Referred specimens.** MPC-D 100/33, partial subadult postcranial skeleton; MPC-D 102/11, partial juvenile skeleton with skull; MPC-D 102/12, adult postcranial skeleton; MPC-D 102/110b, nearly complete juvenile skeleton; MPC-D 102/110c, partial juvenile postcranial skeleton (figures 1–3).

   **Localities and Horizon.** Bugiin Tsav and Guriliin Tsav, Nemegt Basin. Nemegt Formation [28] (lower Maastrichtian).

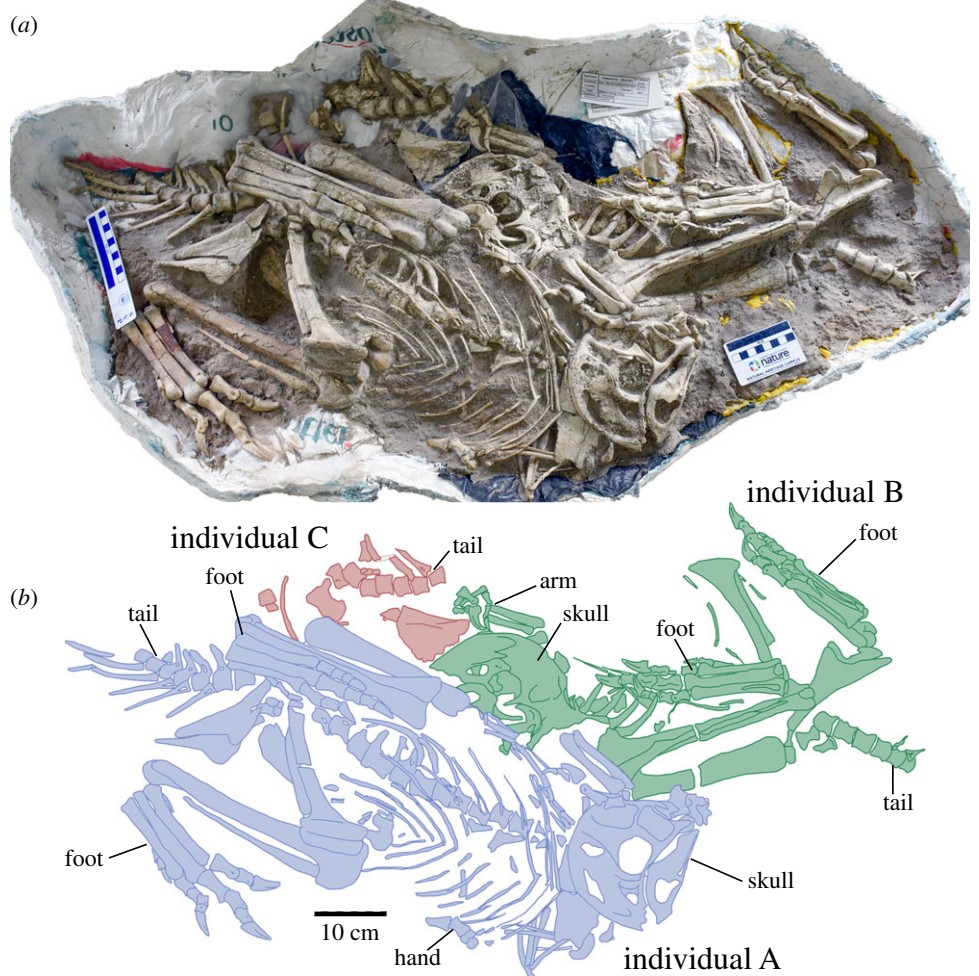

**Figure 1.** Holotype block of *Oksoko avarsan* MPC-D 102/110. (*a,b*) Holotype block with skeletons in ventral view. Colours distinguish different individuals; the holotype individual is in blue.

**Diagnosis.** *Oksoko avarsan* is a small oviraptorid oviraptorosaur distinguished from other oviraptorosaurs by the following suite of autapomorphies (*) and other characters: apically thickened, dome-shaped cranial crest composed equally of nasals and frontals (figure 2)*; nasal recesses housed in a depression; postorbital with dorsally directed frontal process; cervical vertebrae with large epipophyses; functionally didactyl manus (figure 3)*; accessory ridge of brevis fossa of ilium*; anteriorly curving pubis; and large proximodorsal process of distal tarsal IV.

## 2.1. Description

The holotype is one of three articulated juveniles of nearly identical size (see electronic supplementary material; each weighed 44–45 kg based on the method of Campione *et al.* [29]) contained in a single block (MPC-D 102/110; figure 1). Individual A, the holotype, is the most complete, whereas only the right side of individual B and the pelvic region of individual C are preserved. Individuals A and B are crouched in positions that resemble inferred resting poses of other oviraptorids [6,7,30,31], facing opposite directions, with their legs beneath their bodies, arms folded, and heads tucked towards their right arms. Another juvenile skeleton (MPC-D 102/11, 31 kg) was confiscated at the same time, and is preserved in the same crouched posture. Associated with it are the postorbital, quadrate and quadratojugal of a slightly larger individual. These specimens are probably from the same assemblage as the holotype, implying a total of at least four individuals. MPC-D 100/33 was collected in 1974 at Bugiin Tsav, whereas MPC-D 102/12 was collected in 1998 at Guriliin Tsav. Among the known specimens, the entire skeleton of *Oksoko avarsan* is represented (figure 2).

The skull of *Oksoko* (figure 2*b,c*; electronic supplementary material, figures S1 and S2) has a dome-shaped, apically thickened crest composed of the nasals and frontals, with a small contribution from the

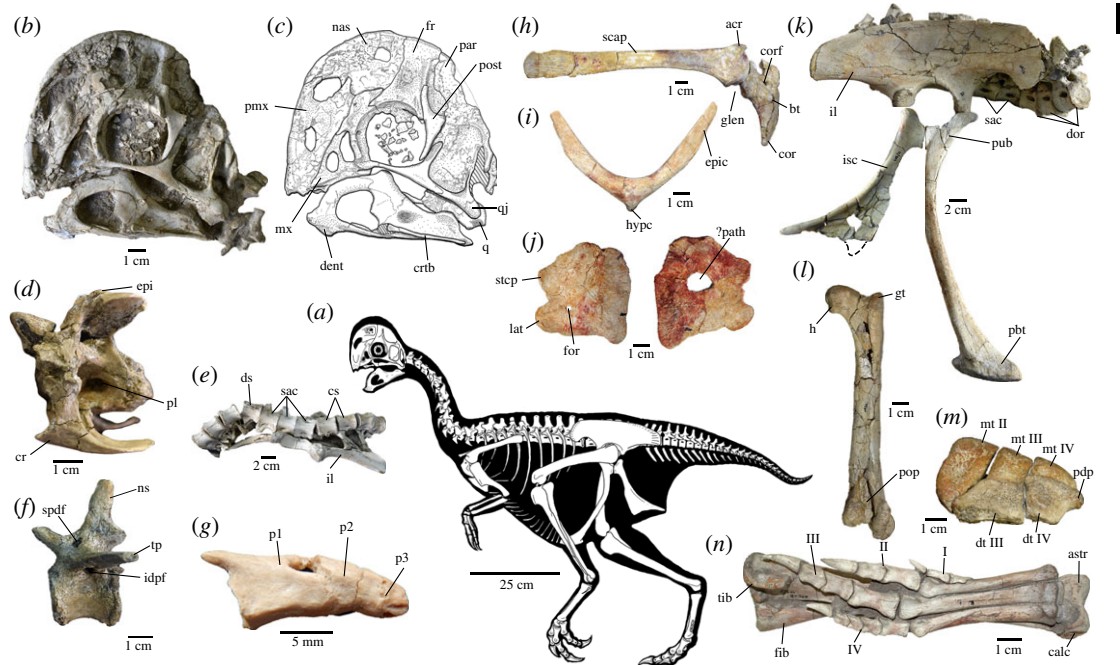

**Figure 2.** Skeletal anatomy of *Oksoko avarsan*. (*a*) Skeletal reconstruction. (*b,c*) Skull of MPC-D 102/110.a in left lateral view. (*d*) Anterior cervical vertebra of MPC-D 102/12 in left lateral view. (*e*) Articulated sacrum and ilium of MPC-D 102/11 in ventral view, anterior is to the left. (*f*) Mid-caudal vertebra of MPC-D 102/12 in left lateral view. (*g*) Pygostyle vertebrae of MPC-D 102/12 in left lateral view. (*h*) Right scapulocoracoid of MPC-D 100/33 in lateral view. (*i*) Furcula of MPC-D 100/33 in anterior view. (*j*) Right and left sternal plates of MPC-D 100/33 in anterior view. (*k*) Articulated pelvis of MPC-D 102/11 in right lateral view. (*l*) Right femur of MPC-D 102/12 in posterior view. (*m*) Proximal metatarsus and distal tarsals of MPC-D 102/12 in proximal view. (*n*) Tibia, fibula and pes of MPC-D 102/11 in ventral view. acr, acromion process; astr, astragalus; bt, biceps tubercle; calc, calcaneum; cor, coracoid; corf, coracoid foramen; cr, cervical rib; crtb, ceratobranchial; cs, caudosacral vertebrae; dent, dentary; dist, distal end; dor, dorsal vertebrae; ds, dorsosacral; dt III, distal tarsal III; dt IV, distal tarsal IV; epi, epipophysis; epic, epicleidum; fib, fibula; for, foramen; fr, frontal; glen, glenoid; gt, greater trochanter; h, head; hypc, hypocleidum; idpf, infradiapophyseal fossa; il, ilium; isc, ischium; I–IV, pedal digits I–IV; lat, lateral trabecula; mt II–IV, metatarsals II–IV; mx, maxilla; nas, nasal; ns, neural spine; p1–p3, pygostyle vertebrae 1–3; par, parietal; ?path, possible pathology; pbt, pubic boot; pdp, proximodorsal process; pl, pleurocoel; pmx, premaxilla; pop, popliteal fossa; post, postorbital; prox, proximal end; pub, pubis; q, quadrate; qj, quadratojugal; sac, sacral vertebrae; scap, scapula; spdf, supradiapophyseal fossa; stcp, sternocoracoidal process; tib, tibia; tp, transverse process.

posteroventrally inclined parietals. The premaxillae are unfused and are laterally depressed below the naris, as in *Citipati* [32]. The lateral process of the nasal has pneumatic recesses set within a shallow depression. The frontal is rare among theropods in being taller than long, a feature shared with *Rinchenia* [33]. A prominent lateral ridge extends dorsally from the postorbital process, which forms the anterior border of the extension of the supratemporal fenestra onto the frontal. The postorbital has parallel jugal and frontal processes, as in *Rinchenia* [33] but unlike all other oviraptorids [1]. The jugal is triradiate and expanded where the rami meet, in contrast with the rod-like jugals of most oviraptorids [1]. The parietal has a flat dorsal surface that tapers transversely in the posterior direction. The interparietal contact lacks a sagittal crest but forms a laterally protruding lip on either side for the attachment of the adductor musculature. The braincase is typical for an oviraptorid [34–36], although the bones that make up this part of the skull remain unfused in MPC-D 102/11. The mandible is like those of most oviraptorids [1], with a pronounced ventral chin and a tall coronoid arch. The ceratobranchial is rod-like and anteriorly expanded, and curves slightly medially. Scleral ossicles are preserved but crushed.

The axis is unusual among those of oviraptorosaurs in having a concave posterior articular surface of the centrum, and the anterior cervical vertebrae have large epipophyses with large lateral pleurocoels (figure 2*d*). The dorsal neural arches are deeply excavated by pneumatic fossae, some of which are coalesced into larger depressions in the posterior dorsals. There are six sacral vertebrae, comprising one dorsosacral, three primordial sacrals and two caudosacrals. Only the three primordial sacrals are fused in MPC-D 102/11 (figure 2*e*). The caudal vertebrae are barrel-shaped and in juveniles have large lateral pleurocoels, which are absent in the adult skeleton (figure 2*f*). A pygostyle composed of

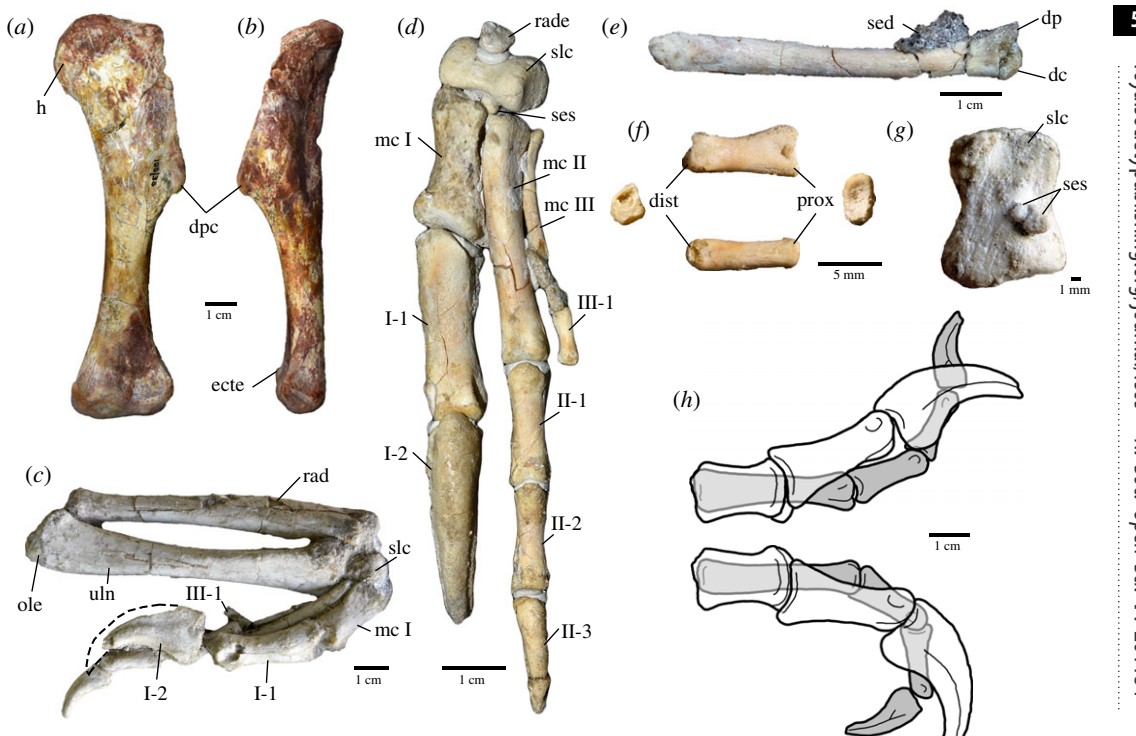

**Figure 3.** Forelimb elements of *Oksoko avarsan*. (*a,b*) Left humerus of MPC-D 100/33 in anterior (*a*) and lateral (*b*) views. (*c*) Right radius, ulna and manus of MPC-D 102/110.a in lateral view. (*d*) Left carpals and manus of MPC-D 102/110.a in dorsal (extensor) view. (*e*) Left metacarpal III of MPC-D 102/110.a in medial view. (*f*) Left manual phalanx III-1 of MPC-D 102/110.a in lateral (top), proximal (right), dorsal (bottom) and distal (left) views. (*g*) Semilunate carpal and sesamoid ossicles of MPC-D 102/110.a in proximal view. (*h*) Range of motion of digits I (white) and II (grey) of *Oksoko avarsan* based on manual manipulation, in full extension (top) and full flexion (bottom). dc, distal condyle; dist, distal end; dp, dorsal process; dpc, deltopectoral crest; ecte, ectepicondylar tuber; ente, entepicondylar tuber; h, head; I-1, manual phalanx I-1; I-2, manual ungual I-2; II-1, manual phalanx II-1; II-2, manual phalanx II-2; II-3, manual ungual II-3; III-1, manual phalanx III-1; mc I–III, metacarpals I–III; ole, olecranon; prox, proximal end; rad, radius; rade, radiale; sed, sediment; ses, sesamoid ossicles; slc, semilunate carpal; uln, ulna. Scale bars as indicated.

three vertebrae is present in MPC-D 102/12 (figure 2*g*), which is inferred to be an adult on the basis of osteohistology (figure 4). However, in MPC-D 102/11, the first pygostyle vertebra appears not to have been fused to the others, which are missing.

The scapulocoracoid (figure 2*h*) is unfused and the glenoid faces posteroventrally. The scapula is long and narrow, with a slightly expanded distal end and a strongly everted acromion. The coracoid has a moderate biceps tubercle and a long posteroventral process. The paired sternal plates are unfused (figure 2*j*), unlike *Heyuannia* [33], and each is wider than long, with a ventrolaterally positioned foramen. The furcula is robust with flat epicleidia and a pointed hypocleidium. The arm is short overall: the combined length of the humerus, ulna and metacarpal II is 109% of femoral length (electronic supplementary material), compared with 112% in *Conchoraptor* (MPC-D 102/03), 128% in *Heyuannia* (HYMV 1–2; MPC-D 100/30) and 162% in *Citipati* (MPC-D 100/42). The humerus, antebrachium and hand are approximately equal in length (figure 3*b,c*). Bivariate plots reveal that the oviraptorosaur forelimb is positively allometric across species (figure 5), which contrasts with the trend of negative allometry in coelurosaurs as a whole [37]. Previous studies have noted this discrepancy [38,39], but the allometric coefficients in those studies were indistinguishable from isometry. The broader sampling of oviraptorosaurs here finds that the allometric coefficient (AC) for the humerus is statistically greater than isometry (AC: 1.14; 95% confidence interval (CI): 1.04–1.23), whereas the ulna (AC: 1.09; CI: 0.97–1.21) and metacarpal II (AC: 1.19; CI: 1.00–1.39) are indistinguishable from isometry. Nonetheless, the forelimb as a whole (humerus + ulna + metacarpal II) is positively allometric (AC: 1.13; CI: 1.01–1.25).

An ovoid radiale articulates with the sellar proximal articular surface of the large semilunate carpal, and two minute ossicles are appressed to the distal face of the latter (figure 3*g*). Based on their positions,

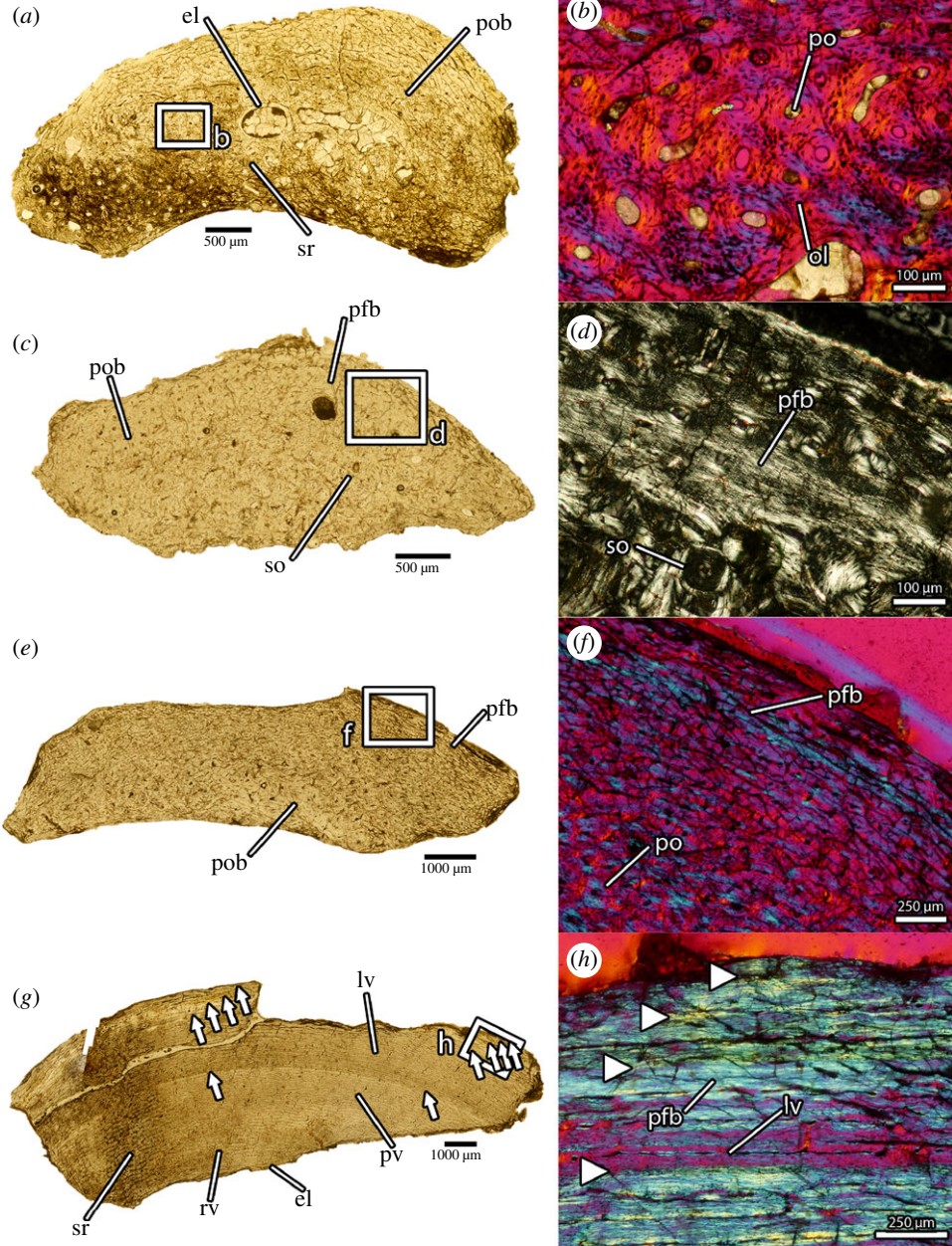

**Figure 4.** Osteohistology of *Oksoko avarsan*. (*a*) Transverse thin section of fibula of MPC-D 102/110.a under normal light. (*b*) Close-up of primary osteonal bone in the cortex of (*a*) under cross-polarized light with lambda filter. (*c*) Transverse thin section of fibula of MPC-D 102/110.b normal light. (*d*) Close-up of annulus of parallel-fibred bone in the cortex of (*c*) under cross-polarized light. (*e*) Transverse thin section of femur of MPC-D 102/11 under normal light. (*f*) Close-up of annulus of parallel-fibred bone in the cortex of (*e*) under cross-polarized light. (*g*) Transverse thin section of femur of MPC-D 102/12 under normal light, showing lines of arrested growth (arrows) and changes in vascularity patterns. (*h*) Close-up of lines of arrested growth (arrows) and parallel-fibred bone in the cortex of (*g*) under cross-polarized light. el, endosteal lamellae; lv, laminar vasculature; ol, osteocyte lacuna; pfb, parallel-fibred bone; po, primary osteon; pob, primary osteonal bone; pv, plexiform vasculature; rv, reticular vasculature; so, secondary osteon; sr, secondary remodelling. Scale bars as indicated.

the two small bones are probably sesamoids, rather than distal carpals, indicating that the carpus comprises only two carpal bones as in other heyuannines [40–42]. To our knowledge, sesamoid bones have not yet been reported in the carpus of theropods, although they are present in some ankylosaurs [43]. Manual digit I is robust (figure 3*d*), with a large trenchant ungual, but is not proportionally longer than in other oviraptorosaurs. The ungual of the gracile second digit is smaller and straighter. Metacarpal III is greatly reduced (figure 3*e*), as in other functionally didactyl theropods [44–46]. A dorsal projection on its distal end would have restricted the movements of the first phalanx

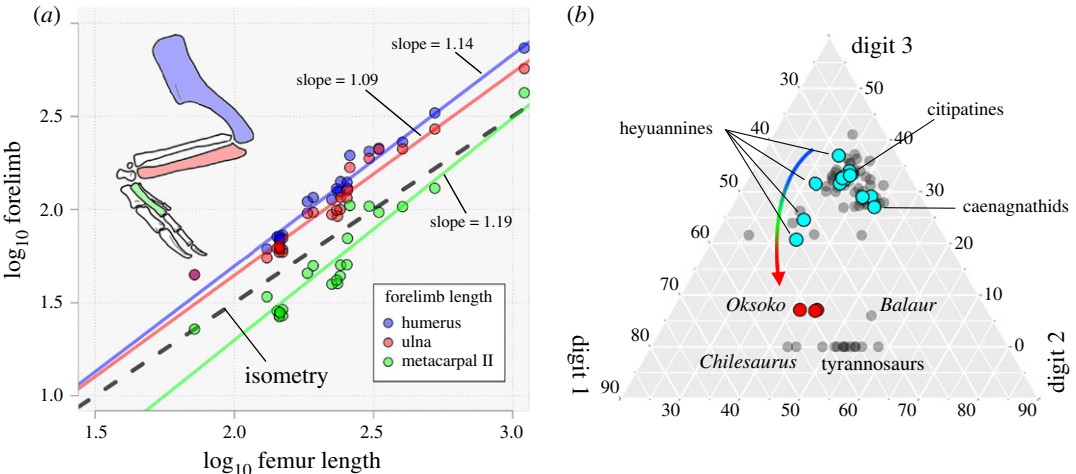

**Figure 5.** Forelimb proportions in Oviraptorosauria. (*a*) Bivariate plot of log-transformed forelimb element length versus log-transformed femoral length, showing slight positive allometry of the forelimb bones to femur length. (*b*) Ternary plot of manual digit proportions of oviraptorosaurs (cyan and red) and other theropods (grey), showing higher disparity in Heyuanninae than other oviraptorosaurs, and trajectory of digit III reduction.

(figure 3*e*). Phalanx III-1 is consistently the only phalanx in the third digit and has a blunted distal end, rather than a distinct condyle (figure 3*f*). It does not extend past the condyle of metacarpal II, so the third digit would not have protruded beyond the metacarpus (figure 3*d*), which suggests that the manus of *Oksoko* would have externally appeared didactyl.

The ilium is long and low with rounded pre- and post-acetabular blades (figure 2*k*), as in most oviraptorids [1]. The brevis fossa has an accessory ridge that is unique to *Oksoko*. The pubes are procurving (figure 2*k*) and share a narrow transverse apron. The unfused ischia are relatively straight and have large obturator processes, distal to which their ventral margins are concave (figure 2*k*). The femur (figure 2*l*) has a well-developed medial head and a low trochanteric ridge. The tibia has a moderately developed cnemial crest, which is smaller than those of caenagnathids [47] but comparable in size to those of other oviraptorids. The distal end of the tibia has an anteromedial flange that cups the unfused astragalocalcaneum. The non-arctometatarsalian pes (figure 2*n*) is unremarkable for an oviraptorid, except that distal tarsal IV bears a bulbous proximodorsal process and distal tarsal III is fused to metatarsal III in adults (figure 2*m*).

## 2.2. Osteohistology

Detailed histological descriptions for each specimen are provided in the electronic supplementary material. The cortices of all specimens are composed predominantly of primary fibrolamellar bone with well-developed osteons, high vascularity and dense osteocyte lacunae (figure 4*b*). The fibulae of MPC-D 102/110a,b and MPC-D 102/11 are consistent in the longitudinal–reticular [48] orientation of vasculature, the development of endosteal lamellae around the small medullary cavities, and the scarcity of secondary remodelling. In each, a zone with a higher proportion of parallel-fibred [49] bone exists towards the outside of the cortex, and this coincides with slightly reduced vascularity (figure 4*d*). Vasculature is otherwise dense throughout the cortex and does not become reduced at the periosteal surface.

The fibula of MPC-D 102/12, by contrast, has lower vascularity, more abundant secondary remodelling, and a much thicker band of parallel-fibred bone at the periosteal surface. At least three lines of arrested growth [48] can be detected in the outer part of the cortex, and it is likely that more have been obliterated by expansion of the medullary cavity and secondary remodelling [50–52]. The femora of MPC-D 102/11 and MPC-D 102/12 contrast starkly in the arrangements of the vasculature and the proportion of parallel-fibred bone. Whereas the cortex of MPC-D 102/11 has reticular vasculature and abundant woven bone [48], especially endosteally, the vasculature of MPC-D 102/12 is arranged into more orderly plexiform–laminar [48] rows, with fewer radial canals (figure 4*g*). The vasculature of the femur of MPC-D 102/12 changes throughout the cortex: both the abundance of radial canals and the density of vasculature overall decrease towards the periosteal surface. These

changes occur in tandem with a transition to parallel-fibred bone with a reduced density of osteocyte lacunae. At least five cyclical growth marks punctuate the femoral cortex of MPC-D 102/12 (figure 4g, h), whereas the femur of MPC-D 102/11 resembles its fibula in having only a faint annulus of parallel-fibred bone near the periosteal surface.

## 3. Discussion

The osteohistology of the specimens suggests that two ontogenetic stages [53] are represented by the material. MPC-D 102/110a,b and MPC-D 102/11 show evidence of rapid growth (well-vascularized [54–56] fibrolamellar bone with dense osteocyte lacunae [57]), but the bone matrix and vasculature of MPC-D 102/12 is more organized, which indicates a slower growth rate [49] (figure 4). This is especially true towards the periosteal surface of the femur of MPC-D 102/12, which shows a transition to parallel-fibred bone with reduced vascularity and closely spaced growth marks [50]. The zones of parallel-fibred bone near the outer cortices of MPC-D 102/110a,b and MPC-D 102/11 are similar to an early growth mark described in the tibia of a caenagnathid [58], and it is likely that they also represent cyclical growth marks. Accordingly, MPC-D 102/110a,b and MPC-D 102/11 are best interpreted as actively growing juveniles at least 1 year old. By contrast, the histology of MPC-D 102/12 is more consistent with an adult that was approaching maximum body size. The presence of five growth marks indicates a minimum age of 5 years, although it is likely that this individual was in fact older and its earlier growth marks were obliterated by expansion of the medullary cavity [50–52]. These ontogenetic stage estimates are supported by patterns of skeletal fusion elsewhere in the body: the braincases, neurocentral sutures and sacral vertebrae of MPC-D 102/110a,b and MPC-D 102/11 are unfused, whereas the distal tarsals, neurocentral sutures and pygostyle of MPC-D 102/12 are fused. The ontogenetic stage of MPC-D 100/33 could not be histologically assessed, but this specimen shows an intermediate degree of fusion: the sacrum is fused, but the distal tarsals and most of the neurocentral sutures are unfused. Accordingly, it was probably intermediate between MPC-D 102/11 and MPC-D 102/12 in ontogenetic stage.

Gregarious behaviour has been inferred in other oviraptorosaurs [59,60], but only unpublished and/ or circumstantial evidence exists for gregariousness in oviraptorids. Although the association of two individuals of the heyuannine *Khaan* suggests that these animals were interacting prior to their deaths [41,61], whether this is evidence of gregarious behaviour is ambiguous. However, the main assemblage of *Oksoko* described here provides, for the first time, clear evidence of gregarious behaviour in oviraptorids. Because the specimens were poached and their exact provenance within the Nemegt Formation is unknown, the sedimentology of the site cannot be assessed. Regardless, some taphonomic information can be gleaned from the skeletons and their arrangement. The crouched posture of the individuals differs from the opisthotonic 'death pose' commonly seen in theropod dinosaurs [62], in which the body lies on one side and the head, neck and tail are arched dorsally. Instead, the feet and belly of each individual are parallel to the bedding plane and the arms and legs are tucked underneath the body—a pose that resembles the resting poses inferred for other non-avian theropods [6,30,31,63–66]. This posture is unlikely to be the result of taphonomy, especially considering that it is consistent in at least three individuals oriented in different directions. Indeed, the pose is nearly identical between MPC-D 102/110.a and MPC-D 102/11, and in both individuals the third toe curves medially to rest medial to the cnemial crest of the tibia (figures 1 and 2; electronic supplementary material, figure S12), a commonality unlikely to be the result of chance. The presence of small, delicate elements like sclerotic plates, and the tight articulation of all the bones, also suggest minimal decay or transport prior to burial. This is further supported by the pristine surface condition of the bones, which argues against extensive scavenging, insect burrowing, or weathering before interment. Thus, we infer that the positions of the skeletons reflect resting postures prior to death and burial, as inferred for other theropods preserved in similar ways [6,7,30,31,63,64,66–68]. However, it is clear that the specimens were dorsoventrally crushed during or after burial, and this has resulted in lateral displacement of the pelvic bones, ribs and possibly other elements of the skeletons. The unexpected positions of the skulls might reflect displacement during burial and compaction, but could also be explained by folding of the neck, as in the sleeping pose of some extant avians [69,70].

Assemblages of multiple articulated skeletons are also known for other theropods including coelophysids [71], ornithomimids [72–74], other oviraptorids [61] and tyrannosaurids [75], as well as for other dinosaurs [76–78], but these assemblages rarely preserve individuals in their inferred resting positions. The excellent preservation of the articulated skeletons suggests that the main *Oksoko*

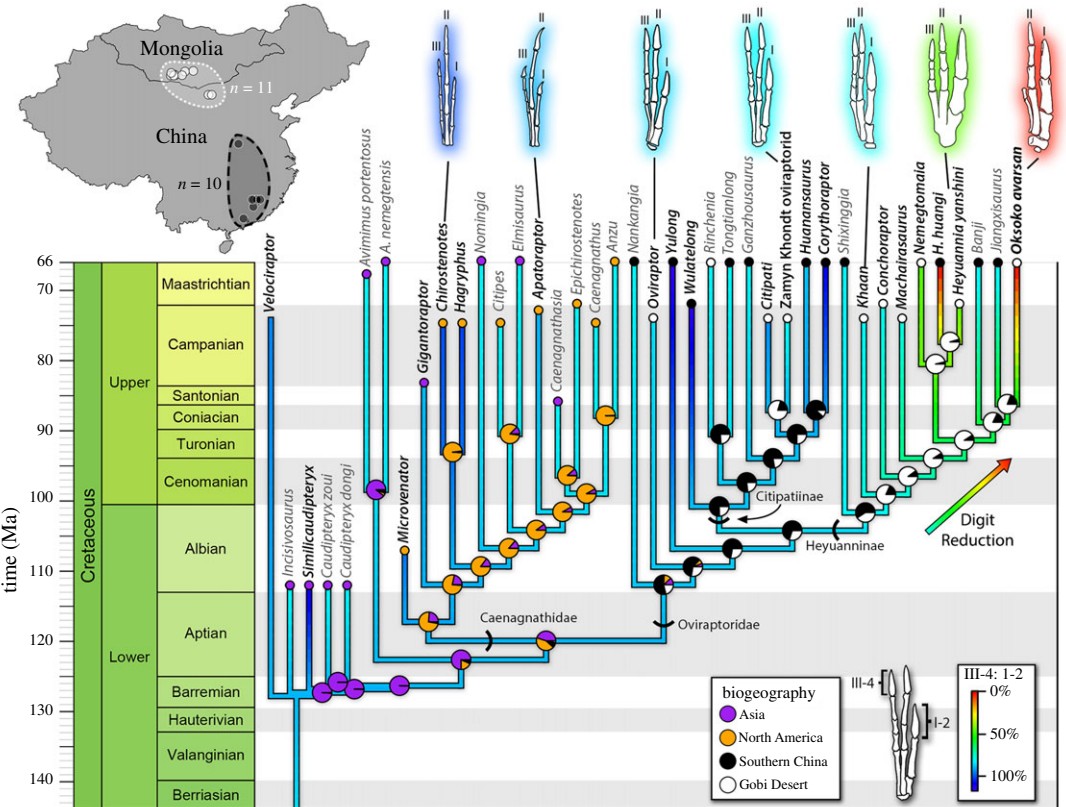

**Figure 6.** Phylogeny, biogeography and digit reduction in Oviraptorosauria. Map of China and Mongolia (top left) showing the distribution of oviraptorids in the Late Cretaceous of Asia. Time-calibrated majority-rule consensus of 9 most parsimonious trees for Oviraptorosauria. Branches are coloured according to maximum-likelihood reconstruction of the ratio of manual ungual III-4 to I-2 (warmer colours indicate a lower ratio), as a proxy for digit reduction. Hand reconstructions show representative morphotypes and increase in robustness of the first digit. Pie charts on nodes represent posterior probability of ancestral state based on stochastic mapping of biogeographic distributions (purple: Asia; orange: North America; black: southern China; white: Gobi Desert). Geographical ranges for each taxon are displayed at the tip of the branch. Taxa with greyed-out names are those for which digit proportion data does not exist, leaving the condition for the entire branch including the tip to be inferred by ancestral state estimation.

assemblage is a mass-death assemblage, rather than a post-mortem aggregation, and that the close association among the individuals is not the result of taphonomic processes. Accordingly, this assemblage provides strong evidence of gregarious behaviour. Like some other theropod assemblages [73], the main *Oksoko* assemblage is comprised solely of juveniles, which suggests that oviraptorid groups were age-segregated. This may have been a by-product of a life-history characterized by multi-year maturation and parental care [79], although no evidence of parental care can be inferred from the main *Oksoko* assemblage. What selective advantages gregarious behaviour conferred on young oviraptorids remains unclear, but possibilities include improved foraging success [80,81], reduced susceptibility to predation [82,83] and increased capacity for thermoregulation [84,85].

Cretaceous faunal interchange between North America and Asia is a well-established fact [86–88], but oviraptorosaur biogeography has traditionally been difficult to interpret because of poor phylogenetic resolution. The greater phylogenetic clarity of our results relative to previous studies allows for reconstructions of biogeography using stochastic mapping and S-DIVA analyses. These show two major range expansions that appear to have led to separate diversification events (figure 6; electronic supplementary material, figure S10). Based on previous work and our analysis, oviraptorosaurs almost certainly originated in Asia [2,89]. Some basal caenagnathoids dispersed to North America by the Albian [90] and gave rise to the caenagnathids, a transition accompanied by changes in the mandible [91] and elongation of the forelimb and manus [92,93]. Oviraptorids were restricted to Asia, and most belong to one of two clusters, centred on the Nanxiong Basin of southern China and the western Gobi Desert of Mongolia (figure 6). When these regions are considered separately, rather than lumped together in a single continent-scale entity, an interesting biogeographic scenario is inferred. In this scenario, the first

oviraptorids originated in southern China, as did Citipatiinae (figure 6). Range expansion as early as the Cenomanian into the western Gobi Desert led to the radiation of Heyuanninae, although taxa continued to disperse between the two regions until at least the Campanian. However, it is important to note that stratigraphic control is poor in both China and Mongolia, and these dispersals may have occurred later than estimated, even as late as the Maastrichtian. In any case, such dispersals must have been rare enough to allow each region to develop its own unique oviraptorid fauna. Range expansion, therefore, was clearly an important factor in the evolution and diversification of Oviraptorosauria as a whole. Dispersals into new regions appear to have precipitated two of the major radiations of oviraptorosaurs: the caenagnathids in North America and the heyuannines in the western Gobi Desert.

The functionally didactyl manus of *Oksoko* is distinctive not only among oviraptorosaurs but also in comparison with other didactyl theropods. Re-examination of *Heyuannia huangi* suggests that this taxon may resemble *Oksoko* in possessing a third digit with only a single, reduced phalanx (electronic supplementary material, figure S13m), a condition that may accordingly have been widespread within Heyuanninae. The third digit in *Oksoko* satisfies the conditions of Senter [94] for a vestigial structure, using other heyuannines with fully developed third digits (e.g. *Khaan*, *Machairasaurus*) and Citipatiinae as outgroups. Typically, digit vestigiality in theropods eventually results in the complete absence of phalanges in the digit in question, as seen in some parvicursorine alvarezsaurs [95,96], *Chilesaurus* [45], *Gualicho* [46] and tyrannosaurids [97]. In this sense, the retention of a single small phalanx in the vestigial digit of *Oksoko* is unusual, but an equivalent condition characterizes the manus of the enigmatic paravian *Balaur* [98,99] and some other non-coelurosaurian theropods like *Carnotaurus*, *Ceratosaurus*, *Coelophysis* and *Tawa* [100–103]. However, the fully developed digits in *Balaur* and *Oksoko* contrast starkly: those of *Balaur* are elongate and well adapted for grasping, whereas those of *Oksoko* are stout and appear to have had reduced ranges of motion (based on manual manipulation of the bones) compared with other oviraptorosaurs [11], which would have limited the grasping ability of the manus (figure 3*h*). Retention of a small vestigial phalanx in the third digit is therefore unlikely to be the result of functional similarity between *Oksoko* and *Balaur*; instead, it probably reflects developmental constraints on the pattern of digit loss in archosaurs [104,105].

To explore trends in oviraptorosaur forelimb evolution, manual proportions and forelimb length were mapped onto a phylogeny using maximum-likelihood ancestral state estimation. This reveals a reduction in the length of both the third digit (figure 6) and the entire forelimb (electronic supplementary material, figure S9) at the base of Heyuanninae, followed by continued reduction in more derived forms. Notably, the onset of forelimb and digit reduction in Heyuanninae appears to coincide with range expansion from southern China into the Gobi Desert (figure 6). The dispersal to the Gobi Desert precedes major changes to the manus (figure 6), which argues against forelimb adaptations for other reasons enabling a dispersal event. Furthermore, the citipatines of the Gobi Desert (*Citipati* and the Zamyn Khondt oviraptorid) also show a slight reduction in the third digit compared with citipatines in southern China (figure 6). Maximum-likelihood reconstruction indicates that this reduction is independent of that in Heyuanninae, but that it also coincides with the arrival of these taxa in the Gobi Desert. This strongly suggests a shift in forelimb function resulting from selection pressures encountered in the new environment. The nature of this niche change is unclear, but it could possibly have been related to diet or foraging style. Early oviraptorosaurs show a trend towards tooth loss that was probably linked to the evolution of herbivory [89,106]. However, the diets of more derived oviraptorosaurs are less certain, because although they show adaptations for herbivory, their edentulous mandibles could have been co-opted for a wide variety of diets [8,9,106]. Direct evidence of diet in oviraptorosaurs is limited to the presence of a gastric mill in *Caudipteryx* [2] and a possible instance of predation in *Oviraptor* [107], which support herbivory and carnivory, respectively. These diets appear to be linked to forelimb and digit reduction and elongation in each of these genera (figure 6; electronic supplementary material, figure S9), although the sample is small. Interestingly, other caudipterids in which the forelimb is not reduced appear to lack a gastric mill [3,108,109], although this could be the result of taphonomy or other factors. Also, the skulls of derived oviraptorosaurs show numerous correlates of herbivory [8,10,106,110], so retention of a plesiomorphic forelimb seemingly well suited to prey capture does not necessarily reflect strict carnivory. In any case, the reduction in length of the forelimbs in Heyuanninae and range of motion of the digits of *Oksoko* imply that these taxa relied less on grasping than other oviraptorosaurs [11] and were probably less well adapted for predatory behaviour [111,112]. Thus, *Oksoko* and other heyuannines were probably more strictly herbivorous than other oviraptorids, but whether herbivory is the primary driver of forelimb reduction in heyuannines is unclear. Indeed, the cranial and mandibular shapes of heyuannines are indistinct from those of citipatines, which suggests minimal differences in diet [110]. Furthermore, at least some

regions of the oviraptorid skull seem to have been subject to strong stabilizing selection [110], suggesting a constrained function. By contrast, the radical departure in the form and evolutionary trend of the heyuannine forelimb suggests that it became functionally decoupled from the skull. Thus, the reduction of the forelimb in heyuannines may be driven by its release from the selective pressures of a foraging function. Instead, the morphology of the forelimb may reflect selection pressures related to nest building, grooming, display or other behaviours, but these are difficult to test. Further study of the biomechanics of the oviraptorosaur forelimb and its evolution may illuminate the drivers of forelimb reduction in heyuannines.

The variation in forelimb length and morphology described here adds to a growing body of evidence of high adaptability in Late Cretaceous oviraptorosaurs [15,33,110,113]. This adaptability may have facilitated the radiation of oviraptorosaurs in the latest Cretaceous. The advent of an edentulous beak and the dietary flexibility it conferred may have enabled caenagnathids to disperse to North America and heyuannines to the Gobi region, resulting in two major diversifications of oviraptorosaurs. Plasticity in forelimb function might have helped oviraptorosaurs disperse to new environments, too, but it may also have aided in coexistence where ranges overlapped. Niche partitioning in oviraptorosaurs has already been suggested on the basis of body size [33,113,114], cranial morphology [110] and mandible morphology [8,9,40]. However, the potential role of the forelimb in niche partitioning has not been previously recognized. Differences in forelimb morphology between caenagnathids and oviraptorids in the Nemegt Basin [33,47,115] may be the result of broader dietary differences already recognized on the basis of the mandibles [110]. Forelimb adaptability could also have contributed to the coexistence of citipatines and heyuannines in the same areas, as the skulls and mandibles of these taxa occupy similar morphospaces [110]. The aberrant bauplan of oviraptorids compared with other theropods [110,116] suggests that they occupied a specialized niche, and variation in the skull, dentary and forelimb may have maximized the available niche space. Oviraptorids were a minor but exceptionally diverse part of the ecosystems they inhabited [14,15,33], and they appear to have been uniquely able to diversify and coexist in the latest Cretaceous ecosystems of Asia.

# 4. Methods

## 4.1. Histological analysis

Thin sections (see electronic supplementary material) were made from the fibular shafts of MPC-D 102/110.a, MPC-D 102/110.b, MPC-D 102/11 and MPC-D 102/12. Additional sections were made from fragments of the femoral shafts of MPC-D 102/11 and MPC-D 102/12. Thin sections were prepared using conventional petrographic methods [49] by embedding shaft fragments in resin, cutting in the appropriate planes, mounting the billets onto slides and polishing to the desired thickness.

## 4.2. Phylogenetic methods

*Oksoko avarsan* was coded into a modified phylogeny [1,14,93,117] and analysed using parsimony (see electronic supplementary material). Character scores were updated using new caenagnathid specimens that improve skeletal representation [114], and some uninformative or poorly constructed characters were removed. The resulting matrix comprised 42 taxa coded for 246 characters and the cladistic analysis was performed in TNT v. 1.1 [118]. Tree searches were run with 10 000 replications of Wagner trees followed by branch swapping using the tree bisection-reconnection algorithm (TBR). A final round of TBR branch swapping was used to find the most parsimonious trees. The analysis recovered nine most parsimonious trees of 641 steps. The strict consensus tree included a polytomy within Caudipteridae and at the base of Citipatiinae, but was otherwise dichotomous. Bremer support for each of the major clades is strong (electronic supplementary material). By contrast, the majority-rule consensus was fully resolved and all clades were recovered in at least 66% of the trees.

The phylogeny was time-scaled using age ranges published in the literature. Although the ages of most taxa could be determined relatively precisely, the stratigraphic ranges of oviraptorids from southern China are poorly constrained. In these cases, stratigraphic ranges were taken from published estimates of the ages of the formations where the specimens were found (electronic supplementary material). Time-scaling was done using the *strap* v. 1.4 package in R v. 3.3.3. Stratigraphic branch lengths were calculated using the equal dating method of Brusatte *et al*. [119].

## 4.3. Statistical methods

Expanded statistical methods are outlined in the electronic supplementary material. The length ratio of manual ungual III-4 to manual ungual I-2 was mapped as a quantitative character onto a phylogeny scaled using time-calibrated branch lengths and maximum likelihood to estimate ancestral states and missing tips using the *phytools* package in R. To more accurately constrain the root condition, additional outgroups representing a broader array of coelurosaurs were grafted to the preferred tree (following the topology in Hendrickx *et al.* [120]) and included in all subsequent analyses. The resulting ancestral state estimation was visualized with warm colours indicating a low ratio of III-4 : I-2 (i.e. smaller third digit), and cool colours representing a higher ratio of III-4 : I-2 (i.e. larger third digit).

Palaeobiogeography was examined by creating discrete bins that were analysed as a categorical character. To make these bins more informative, they were chosen based on the biogeographic transition of interest in the relevant part of the phylogeny. For example, to understand the dispersal of caenagnathids into North America, basal oviraptorosaurs and caenagnathids were coded as either 'Asian' (purple) or 'North American' (orange), without subdividing either of those regions. Because all oviraptorids are found in Asia, two subdivisions representing the main basins were created: southern China (black) and Gobi Desert (white). Biogeographic histories were stochastically simulated to estimate ancestral states. Biogeographic estimations were integrated with the digit reduction data by plotting biogeographic ancestral state likelihoods onto the nodes of a tree on which digit reduction was mapped as a continuous character.

Ethics. Material described in this manuscript includes both legitimately collected (MPC-D 100/33, MPC-D 102/12) and unlawfully collected specimens (MPC-D 102/11, MPC-D 102/110) that were confiscated before leaving the country by Mongolian authorities in December 2006. More details about the history of the specimens are outlined in the electronic supplementary material. The legitimately collected specimens were collected under all appropriate permits to the Institute of Paleontology, Mongolian Academy of Sciences, Ulaanbaatar, Mongolia. All specimens are now accessioned at the Institute of Paleontology, Mongolian Academy of Sciences, Ulaanbaatar, Mongolia, where they are available for study. Therefore, all of the specimens adhere to the Society of Vertebrate Paleontology's guidelines that 'Scientifically significant fossil vertebrate specimens, along with ancillary data, should be curated and accessioned in the collections of repositories charged in perpetuity with conserving fossil vertebrates for scientific study and education (e.g. accredited museums, universities, colleges and other educational institutions)'. We endorse this bylaw and we do not condone the unlawful collection of fossil resources, which resulted in a loss of available information for this study.

Data accessibility. All data and code used in the study is either provided in the electronic supplementary material, or deposited at Dryad (https://doi.org/10.5061/dryad.31zcrjdhq) [121]. TNT software is available at https://cladistics.org/tnt/; R software is available at https://www.r-project.org; RASP is available at http://mnh.scu.edu.cn/soft/blog/RASP. This published work and the nomenclatural acts it contains have been registered in ZooBank, the proposed online registration system for the International Code of Zoological Nomenclature. The ZooBank life science identifiers (LSID) can be resolved and the associated information viewed by appending the LSID to the prefix http://zoobank.org/. The LSID for this publication is: urn:lsid:zoobank.org:act:BC156FC2-D79A-48CD-BF27-8B1EEC1ABF78.

Authors' contributions. G.F.F. designed the study, collected data, performed the comparative and analytical work, and wrote the paper. T.C, K.T., Y.K. and C.S. provided data and contributed to writing and discussion. P.J.C. collected data, performed analyses and helped write the paper.

Competing interests. The authors declare no competing interests.

Funding. G.F.F. is funded by the Royal Society (grant no. NIF\R1\191527), Vanier Canada, NSERC, Alberta Innovates and the Dinosaur Research Institute. C.S. is funded by NSERC (grant no. RGPIN 2017-06246) and start-up funding awarded by the University of Alberta. P.J.C. is funded by NSERC (grant no. RGPIN 2017-04715).

Acknowledgements. We thank S. Ulziitseren for access to specimens and C. Bayardorj for preparation of MPC-D 102/11 and MPC-D 102/12. We thank S. Brusatte for helpful discussion. We thank the numerous reviewers for their comments on earlier drafts of this manuscript, which greatly improved the study.

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
