## [Reviewer comments · Royal Society Open Science]

Review History

RSOS-201184.R0 (Original submission)

Review form: Reviewer 1 (Alexander Vargas)

Is the manuscript scientifically sound in its present form?

Yes

Are the interpretations and conclusions justified by the results?

Yes

Is the language acceptable?

Yes

Do you have any ethical concerns with this paper?

No

Have you any concerns about statistical analyses in this paper?

No

Recommendation?

Accept with minor revision (please list in comments)

Comments to the Author(s)

This is a significant contribution that clarifies the phylogeny, sociality and biogeographical history of Oviraptorosauria, uncovering an interesting correspondence between the radiation of Heyuannine oviraptorosaurs into the Gobi region and their reduced forelimbs, which suggest it happened along new ecological specialization. The forelimbs are interesting not only because of reduction but also because of the loss of a functional digit 2. I am guessing my selection as a reviewer is related to my research in the evolution of theropod forelimbs. My comments will be mostly focused on this aspect as I am not an expert on Oviraptorosauria: I cannot comment much on the description, diagnosis, or character matrix used for the phylogenetic analysis.

I have only minor revisions; in general, the manuscript seems sound.

My most obvious correction would be that the legend in figure 5 says "showing slight negative allometry of the forelimb to femur length". Surely the authors meant "slight positive allometry" (as can be seen in the graph, and as mentioned in the main text). The fact that Oviraptorosauria do not show negative allometry (unlike other theropods) has been recently mentioned by my lab in Palma-Liberona et al 2019, and previously, by Lü et al 2013 (perhaps also, Xu et al 2018). It would be scholarly to include these references as well as the allometric coefficient (slope), maybe briefly in parenthesis when mentioning the slight positive allometry, hopefully with a confidence interval as well. This is because allometric coefficients for Oviraptorosauria were found to be only very slightly greater than 1 in these previous works, and statistically indistinguishable from isometry. Since one of the main figures and analyses of this paper are devoted to this, and other researchers are likely interested, it is worth citing these works and clarifying these matters, specially if previous assessments are being corrected by further/better data.

I agree with the authors that a single vestigial phalanx on the reduced digit III reflects developmental trends. Something similar can be observed with reduction of digit IV in the manus of *Coelophysis*, and reduction of digit V in *Tawa* (a fact the authors may, or may not, choose to mention). I also agree that decreased function is the right explanation for reduction.

Regarding phylogenetic statistics, the authors may want to use some method that allows better treatment of uncertainty in geological age for the estimation of branch length. This could actually be a more serious issue considering the fact that the ages from China appear to be uncertain in a rather wide range. A way to handle this is presented in Palma-Liberona et al. 2019 using the cal3 algorithm of the R package paleotree v2.7.

I hope these comments are useful. References are included below.

Sincerely,
A. Vargas

Refs:

Lü J, Currie PJ, Xu L, Zhang X, Pu H, Jia S. Chicken-sized oviraptorid dinosaurs from Central China and their ontogenetic implications. *Naturwissenschaften*. 2013;100:165-75.

Palma Liberona, J.A., Soto-Acuña, S., Mendez, M.A. et al. Assessment and interpretation of negative forelimb allometry in the evolution of non-avian Theropoda. *Front Zool* 16, 44 (2019). <https://doi.org/10.1186/s12983-019-0342-9>

Xu, X., Choiniere, J., Tan, Q., Benson, R. B., Clark, J., Sullivan, C., ... & Wang, S. (2018). Two Early Cretaceous fossils document transitional stages in alvarezsaurian dinosaur evolution. *Current Biology*, 28(17), 2853-2860.

Review form: Reviewer 2

Is the manuscript scientifically sound in its present form?

Yes

Are the interpretations and conclusions justified by the results?

Yes

Is the language acceptable?

Yes

Do you have any ethical concerns with this paper?

No

Have you any concerns about statistical analyses in this paper?

No

Recommendation?

Accept with minor revision (please list in comments)

Comments to the Author(s)

I believe this is a good paper describing important specimens. The followings are questions on the content that I feel need to be answered:

Line 97, designation of the holotype: I am against designating a poached specimen as the holotype if it can be avoided. The primary reason is an ethical one - obviously the authors know it too well, but poaching should be discouraged at all cost and designating a poached specimen the name-bearer could send a wrong message. Also, no matter how well the formation or locality of such a specimen can be constrained, it will never be confirmed for sure. Because a couple of referred specimens listed here are with concrete locality information, I believe either of those specimens should be the holotype so long as they preserve diagnostic features of the species despite the fact that they may be less complete than the poached ones.

Line 112, diagnosis: I may be wrong, but listing non-arctometatarsalin pes is appropriate?

Line 148: This is minor. Is the "articular surface" that of the centrum? If so, please specify as such.

Line 343: "Reduction of the third digit" depicted by the ratio between the unguals of digits I and III could instead reflect hyper development of the first ungual. I doubt this ratio as a good parameter for indicating the degree of reduction of the third digit.

Line 345: To me, the "trend" of the digit reduction shown in Fig. 6 appears to be created by two outliers, Oksoko and H. yanshini. Other heyuannines seem to have similar values, not successively smaller ratios. Is this really a trend?

Line 348: You need to provide more circumstantial evidence or more detailed functional explanations to propose any causal relationship between geographic and morphological shifts. Otherwise, it is not convincing.

Line 364: Could you provided a range of motion estimate of a "typical" oviraptorosaur so we can compare and see that this species "relied less on grasping"?

Decision letter (RSOS-201184.R0)

Dear Dr Funston

On behalf of the Editors, we are pleased to inform you that your Manuscript RSOS-201184 "A new two-fingered dinosaur sheds light on the radiation of Oviraptorosauria" has been accepted for publication in Royal Society Open Science subject to minor revision in accordance with the referees' reports. Please find the referees' comments along with any feedback from the Editors below my signature.

(The Associate Editor has made the comment: "The two reviews complement each other very well.

Concerning what specimens to use as the holotype, one of the reviewers made a sensible recommendation which I think the authors should consider, but I have no problem if they stick to their original plan.")

Please submit your revised manuscript and required files (see below) no later than 7 days from today's (ie 03-Sep-2020) date. Note: the ScholarOne system will 'lock' if submission of the revision is attempted 7 or more days after the deadline. If you do not think you will be able to meet this deadline please contact the editorial office immediately.

on behalf of Professor Marcelo Sanchez (Associate Editor) and Peter Haynes (Subject Editor)
openscience@royalsociety.org

Reviewer comments to Author:
Reviewer: 1

Comments to the Author(s)

This is a significant contribution that clarifies the phylogeny, sociality and biogeographical history of Oviraptorosauria, uncovering an interesting correspondence between the radiation of

Heyuannine oviraptorosaurs into the Gobi region and their reduced forelimbs, which suggest it happened along new ecological specialization. The forelimbs are interesting not only because of reduction but also because of the loss of a functional digit 2. I am guessing my selection as a reviewer is related to my research in the evolution of theropod forelimbs. My comments will be mostly focused on this aspect as I am not an expert on Oviraptorosauria: I cannot comment much on the description, diagnosis, or character matrix used for the phylogenetic analysis.

I have only minor revisions; in general, the manuscript seems sound.

My most obvious correction would be that the legend in figure 5 says "showing slight negative allometry of the forelimb to femur length". Surely the authors meant "slight positive allometry" (as can be seen in the graph, and as mentioned in the main text). The fact that Oviraptorosauria do not show negative allometry (unlike other theropods) has been recently mentioned by my lab in Palma-Liberona et al 2019, and previously, by Lü et al 2013 (perhaps also, Xu et al 2018). It would be scholarly to include these references as well as the allometric coefficient (slope), maybe briefly in parenthesis when mentioning the slight positive allometry, hopefully with a confidence interval as well. This is because allometric coefficients for Oviraptorosauria were found to be only very slightly greater than 1 in these previous works, and statistically indistinguishable from isometry. Since one of the main figures and analyses of this paper are devoted to this, and other researchers are likely interested, it is worth citing these works and clarifying these matters, specially if previous assessments are being corrected by further/better data.

I agree with the authors that a single vestigial phalanx on the reduced digit III reflects developmental trends. Something similar can be observed with reduction of digit IV in the manus of *Coelophysis*, and reduction of digit V in *Tawa* (a fact the authors may, or may not, choose to mention). I also agree that decreased function is the right explanation for reduction.

Regarding phylogenetic statistics, the authors may want to use some method that allows better treatment of uncertainty in geological age for the estimation of branch length. This could actually be a more serious issue considering the fact that the ages from China appear to be uncertain in a rather wide range. A way to handle this is presented in Palma-Liberona et al. 2019 using the cal3 algorithm of the R package paleotree v2.7.

I hope these comments are useful. References are included below.

Sincerely,
A. Vargas

Refs:

Lü J, Currie PJ, Xu L, Zhang X, Pu H, Jia S. Chicken-sized oviraptorid dinosaurs from Central China and their ontogenetic implications. *Naturwissenschaften*. 2013;100:165–75.

Palma Liberona, J.A., Soto-Acuña, S., Mendez, M.A. et al. Assessment and interpretation of negative forelimb allometry in the evolution of non-avian Theropoda. *Front Zool* 16, 44 (2019). <https://doi.org/10.1186/s12983-019-0342-9>

Xu, X., Choiniere, J., Tan, Q., Benson, R. B., Clark, J., Sullivan, C., ... & Wang, S. (2018). Two Early Cretaceous fossils document transitional stages in alvarezsaurian dinosaur evolution. *Current Biology*, 28(17), 2853-2860.

Reviewer: 2

Comments to the Author(s)

I believe this is a good paper describing important specimens. The followings are questions on the content that I feel need to be answered:

Line 97, designation of the holotype: I am against designating a poached specimen as the holotype if it can be avoided. The primary reason is an ethical one - obviously the authors know it too well, but poaching should be discouraged at all cost and designating a poached specimen the name-bearer could send a wrong message. Also, no matter how well the formation or locality of such a specimen can be constrained, it will never be confirmed for sure. Because a couple of referred specimens listed here are with concrete locality information, I believe either of those specimens should be the holotype so long as they preserve diagnostic features of the species despite the fact that they may be less complete than the poached ones.

Line 112, diagnosis: I may be wrong, but listing non-arctometatarsalin pes is appropriate?

Line 148: This is minor. Is the "articular surface" that of the centrum? If so, please specify as such.

Line 343: "Reduction of the third digit" depicted by the ratio between the unguals of digits I and III could instead reflect hyper development of the first unguual. I doubt this ratio as a good parameter for indicating the degree of reduction of the third digit.

Line 345: To me, the "trend" of the digit reduction shown in Fig. 6 appears to be created by two outliers, Oksoko and H. yanshini. Other heyuannines seem to have similar values, not successively smaller ratios. Is this really a trend?

Line 348: You need to provide more circumstantial evidence or more detailed functional explanations to propose any causal relationship between geographic and morphological shifts. Otherwise, it is not convincing.

Line 364: Could you provided a range of motion estimate of a "typical" oviraptorosaur so we can compare and see that this species "relied less on grasping"?

===PREPARING YOUR MANUSCRIPT===

If you have been asked to revise the written English in your submission as a condition of publication, you must do so, and you are expected to provide evidence that you have received language editing support. The journal would prefer that you use a professional language editing service and provide a certificate of editing, but a signed letter from a colleague who is a native speaker of English is acceptable. Note the journal has arranged a number of discounts for authors

using professional language editing services
(<https://royalsociety.org/journals/authors/benefits/language-editing/>).

===PREPARING YOUR REVISION IN SCHOLARONE===

Author's Response to Decision Letter for (RSOS-201184.R0)

See Appendix A.

Decision letter (RSOS-201184.R1)

Dear Dr Funston,

It is a pleasure to accept your manuscript entitled "A new two-fingered dinosaur sheds light on the radiation of Oviraptorosauria" in its current form for publication in Royal Society Open Science.

on behalf of Professor Marcelo Sanchez (Associate Editor) and Peter Haynes (Subject Editor)
openscience@royalsociety.org

Appendix A

Royal Society Open Science
Manuscript RSOS-201184

Dear Dr. Dunn,

We thank you and the reviewers for your time and effort in assessing our manuscript. We find the reviews constructive and we believe their incorporation into the paper has strengthened it. In addition to the point-by-point comments below, I have detailed the changes to the manuscript in a marked-up document using the 'Track Changes' feature of Microsoft Word.

General changes:

- We have added the statistics for the regressions to the figure and the text, as per the suggestion of reviewer 1.
- We have strengthened the discussion of the relationship between geographic and morphological shifts, providing more evidence in favour of our argument that the dispersal resulted in a shift in function.
- We have added extra references to other theropods, in the context of digit reduction and range of motion/function.

Sincerely,

Gregory F. Funston, PhD
Royal Society Newton International Fellow
University of Edinburgh
Edinburgh, UK
Gregory.Funston@ed.ac.uk

Reviewer comments to Author:

Reviewer: 1

Comments to the Author(s)

This is a significant contribution that clarifies the phylogeny, sociality and biogeographical history of Oviraptorosauria, uncovering an interesting correspondence between the radiation of Heyuannine oviraptorosaurs into the Gobi region and their reduced forelimbs, which suggest it happened along new ecological specialization. The forelimbs are interesting not only because of reduction but also because of the loss of a functional digit 2. I am guessing my selection as a reviewer is related to my research in the evolution of theropod forelimbs. My comments will be mostly focused on this aspect as I am not an expert on Oviraptorosauria: I cannot comment much on the description, diagnosis, or character matrix used for the phylogenetic analysis.

Thank you for your positivity regarding our manuscript. We appreciate your insights on forelimb evolution.

I have only minor revisions; in general, the manuscript seems sound.

My most obvious correction would be that the legend in figure 5 says "showing slight negative allometry"

of the forelimb to femur length". Surely the authors meant "slight positive allometry" (as can be seen in the graph, and as mentioned in the main text).

Yes, absolutely correct, that is an error in the figure caption. It has been changed to reflect the positive allometry we recovered. Thanks for catching that!

The fact that Oviraptorosauria do not show negative allometry (unlike other theropods) has been recently mentioned by my lab in Palma-Liberona et al 2019, and previously, by Lü et al 2013 (perhaps also, Xu et al 2018). It would be scholarly to include these references as well as the allometric coefficient (slope), maybe briefly in parenthesis when mentioning the slight positive allometry, hopefully with a confidence interval as well. This is because allometric coefficients for Oviraptorosauria were found to be only very slightly greater than 1 in these previous works, and statistically indistinguishable from isometry. Since one of the main figures and analyses of this paper are devoted to this, and other researchers are likely interested, it is worth citing these works and clarifying these matters, specially if previous assessments are being corrected by further/better data.

We have added the slope and 95% confidence interval to the lines in Figure 5, and we have added references to Palma-Liberona et al 2019, of which we were not aware. This is a valuable resource, so thank you for sharing it. We have also added references to Lü et al. (2013), but in reviewing Xu et al. 2018 we could not find any references to allometric coefficients in theropod subclades other than alvarezsaurids.

I agree with the authors that a single vestigial phalanx on the reduced digit III reflects developmental trends. Something similar can be observed with reduction of digit IV in the manus of Coelophysis, and reduction of digit V in Tawa (a fact the authors may, or may not, choose to mention). I also agree that decreased function is the right explanation for reduction.

We have added in references to the retention of a single phalanx in *Coelophysis* and *Tawa*.

Regarding phylogenetic statistics, the authors may want to use some method that allows better treatment of uncertainty in geological age for the estimation of branch length. This could actually be a more serious issue considering the fact that the ages from China appear to be uncertain in a rather wide range. A way to handle this is presented in Palma-Liberona et al. 2019 using the cal3 algorithm of the R package paleotree v2.7.

The uncertainty in geological age for Asian oviraptorosaurs is certainly troublesome, and little work has examined the geology of the Henan Basin, where many of the Chinese specimens are found. Likewise, the absence of datable volcanics or palynomorphs in the Gobi of Mongolia means these date ranges cannot be further constrained at the present. Nonetheless, we feel that this has minimal impact on the study because we do not attempt to estimate the rate of evolution of these characters, we are simply showing a trend. The lengths of the branches have minimal impact on the presence of the trend, but they could affect the perceived strength of that trend. However, we believe we have been conservative in this respect because by including a broader temporal range for each taxon, we increase branch length. This, in turn, reduces the ease of perception of a trend. Early sensitivity tests we ran using different methods of branch scaling had minimal effects on the presence of a trend, and in fact most of these produced a more pronounced trend than the one we show, because they had shorter branch lengths. Regardless, we chose to use the 'equal' method because it is widely accepted. Although it's possible that the cal3 method could produce tighter node constraints, the limited scope of our study means that we

can't use birth/extinction/sampling rates derived for larger theropod datasets. In our opinion, determining these is out of the scope of the study. Altogether, we believe that although not perfect, our method is conservative, and, if anything, would reduce the perceived strength of the trend, rather than artificially create one.

I hope these comments are useful. References are included below.

Sincerely,
A. Vargas

Reviewer: 2

Comments to the Author(s)

I believe this is a good paper describing important specimens. The followings are questions on the content that I feel need to be answered:

Thank you for your positivity regarding our manuscript.

Line 97, designation of the holotype: I am against designating a poached specimen as the holotype if it can be avoided. The primary reason is an ethical one - obviously the authors know it too well, but poaching should be discouraged at all cost and designating a poached specimen the name-bearer could send a wrong message. Also, no matter how well the formation or locality of such a specimen can be constrained, it will never be confirmed for sure. Because a couple of referred specimens listed here are with concrete locality information, I believe either of those specimens should be the holotype so long as they preserve diagnostic features of the species despite the fact that they may be less complete than the poached ones.

We certainly agree in principle, although in this situation we believe our approach is justified. The specimens were poached, but unlike many other poached specimens (e.g. *Deinocheirus*, *Halszkaraptor*, and the new sauropod embryonic skull from Argentina), MPC-D 102/110 never left the country. It was halted and confiscated at the border, and therefore its provenance from within Mongolia is certain. Geochemical testing has helped to narrow down a possible locality, but, like you say, we may never know for certain the site that produced the specimens (although this has happened—look at *Deinocheirus*). However, this is also true for many historic specimens, even those legitimately collected, that pre-dated GPS positioning or detailed maps. Considering that the provenance of a taxon is not part of its diagnosis, and that the holotype serves to represent the diagnostic characters of a taxon, we believe we are justified in using this specimen as the holotype. It is the most complete specimen, and it provides an exemplary comparison for future studies. Seeing as the specimen is publicly accessioned at the Institute of Palaeontology in Ulaanbaatar, it satisfies SVP's guidelines on ethical use of specimens. We agree that poaching should be discouraged and many of us on the paper are vocal about this in palaeontological circles (PJC especially). We hope that the specific epithet: *avarsan*, meaning 'rescued', will impart the proper message that poaching is detrimental, rather than glorifying it.

Line 112, diagnosis: I may be wrong, but listing non-arctometatarsalin pes is appropriate?

We initially included this character because it serves to distinguish *Oksoko* as an oviraptorid, but we have removed it because it is redundant.

Line 148: This is minor. Is the "articular surface" that of the centrum? If so, please specify as such.

Yes, it is. We have added a statement to that effect.

Line 343: "Reduction of the third digit" depicted by the ratio between the unguis of digits I and III could instead reflect hyper development of the first unguis. I doubt this ratio as a good parameter for indicating the degree of reduction of the third digit.

As described in the methods and supplement, this ratio was chosen to maximize the available data for phylogenetic analysis. While it likely does partly reflect an increase in the size of the first digit, this increase is more in robustness than in length, based on our extensive personal observations. This is clear from our Figure 5b, which shows in the ternary plot that the increase in length of digit I relative to digit II (the horizontal axis) is much less profound than the reduction of digit III relative to the other digits (the vertical axis). Indeed, it is difficult to conceive of why the functional third digit would be lost in *Heyuannia* and *Oksoko* as a result of hyper-development of the first digit. Although ideally we would compare digit III to an independent portion of the body (e.g. femur), the fossil record of oviraptorosaurs is biased and the data does not currently exist to do so.

Line 345: To me, the "trend" of the digit reduction shown in Fig. 6 appears to be created by two outliers, Oksoko and H. yanshini. Other heyuannines seem to have similar values, not successively smaller ratios. Is this really a trend?

This impression is partly created by missing data, which we have indicated by bold (present) and non-bold (missing) names. The two taxa most closely related to *Oksoko* have missing data, and so the maximum likelihood method predicts these tips as the mean value. This is also partly obfuscated by the limits of the gradient we used. Because one of the limits is zero, it means that most of the measurements fall towards the middle of the gradient, where colours are more difficult to distinguish. We tried other colour schemes to try to deal with this, but this one was the most effective. The trend becomes more clear looking at the nodes rather than the tips themselves.

Line 348: You need to provide more circumstantial evidence or more detailed functional explanations to propose any causal relationship between geographic and morphological shifts. Otherwise, it is not convincing.

We have provided more evidence that the geographic and morphological shifts are linked. In particular, we highlight that dispersal occurred prior to major morphological changes, and therefore changes to the forelimb for other reasons did not enable a dispersal. Similarly, we note that citipatines inhabiting the Gobi Desert have independently reduced third digits compared to their relatives, which also coincide with dispersal to the Gobi. It thus appears that the Gobi's digit-reducing effect also applied to citipatines, although they dispersed later, which may be why it is less pronounced. We have already highlighted the need for more biomechanical inquiry into the oviraptorosaur forelimb (which would be comprehensive studies on their own), and hopefully this study leads to more interest in the subject.

Line 364: Could you provided a range of motion estimate of a "typical" oviraptorosaur so we can compare and see that this species "relied less on grasping"?

We have added a reference to Senter (2005), which describes range of motion in *Chirostenotes*, a caenagnathid. We have also added references to other theropods with ranges of motion available in the literature.